# Nutrient content and stoichiometry of pelagic *Sargassum* reflects increasing nitrogen availability in the Atlantic Basin

B. E. Lapointe [1✉], R. A. Brewton[1], L. W. Herren [1], M. Wang [2], C. Hu [2], D. J. McGillicuddy Jr. [3], S. Lindell [3], F. J. Hernandez [4] & P. L. Morton [5]

The pelagic brown macroalgae *Sargassum* spp. have grown for centuries in oligotrophic waters of the North Atlantic Ocean supported by natural nutrient sources, such as excretions from associated fishes and invertebrates, upwelling, and $N_2$ fixation. Using a unique historical baseline, we show that since the 1980s the tissue %N of *Sargassum* spp. has increased by 35%, while %P has decreased by 44%, resulting in a 111% increase in the N:P ratio (13:1 to 28:1) and increased P limitation. The highest %N and $\delta^{15}$N values occurred in coastal waters influenced by N-rich terrestrial runoff, while lower C:N and C:P ratios occurred in winter and spring during peak river discharges. These findings suggest that increased N availability is supporting blooms of *Sargassum* and turning a critical nursery habitat into harmful algal blooms with catastrophic impacts on coastal ecosystems, economies, and human health.

[1] Harbor Branch Oceanographic Institute, Florida Atlantic University, Fort Pierce, FL, USA. [2] College of Marine Science, University of South Florida, St. Petersburg, FL, USA. [3] Woods Hole Oceanographic Institution, Woods Hole, MA, USA. [4] Division of Coastal Sciences, University of Southern Mississippi, Ocean Springs, MS, USA. [5] Florida State University/National High Magnetic Field Lab, Tallahassee, FL, USA. ✉email: blapoin1@fau.edu

For over five centuries, the floating brown macroalgae of the North Atlantic Ocean (NA) known as pelagic *Sargassum* has stirred debate and mystery among seafarers and scientists alike. This vegetation was first described by Christopher Columbus and his sailors in 1492, which reminded them of "salgazo," small grapes in Portugal, and thus the name of the central gyre of the NA became the Sargasso Sea[1]. The vegetation is comprised of two holopelagic *Sargassum* species, *S. natans* and *S. fluitans*, that reproduce solely by vegetative propagation[2]. Early oceanographers and marine botanists thought this vegetation grew primarily in the Sargasso Sea, which they estimated to contain 7 to 10 million tons[3,4] (Fig. 1). However, this presented a paradox to modern oceanographers who considered the Sargasso Sea a biological desert due to the very low nutrient concentrations and biological productivity in its surface waters (Ryther's Paradox)[1].

This paradox has since been explained by the seasonal transport of nutrient enriched and productive *Sargassum* from the Gulf of Mexico (GOM), Loop Current, and Gulf Stream to the Sargasso Sea. Studies of the productivity and nutrition of pelagic *Sargassum* showed that neritic plants in the southeastern GOM, Loop Current, and western wall of the Gulf Stream along the southeastern United States had twofold higher productivity and lower carbon:nitrogen (C:N) and carbon:phosphorus (C:P) ratios compared to oceanic populations in the Sargasso Sea[5,6]. Major advances in remote sensing of *Sargassum* using Medium Resolution Imaging Spectrometer (MERIS) and Moderate Resolution Imaging Spectroradiometer (MODIS) satellite imagery revealed extensive and frequent windrows of *Sargassum* (line-shaped aggregations formed by wind forcing) in the western GOM in 2004 and 2005[7]. High biomass strandings of *Sargassum* along GOM coastlines since the 1980s have led to intensive beach raking[8] and an emergency shutdown of a nuclear power plant on the west coast of Florida[9], perhaps as a consequence of increasing N inputs to the GOM from the Mississippi River and its distributary the Atchafalaya River, as well as other land-based sources[5,10]. The extensive biomass of *Sargassum* in the western

GOM is proposed to be advected seasonally via the Loop Current and Gulf Stream to the Sargasso Sea[11]. For the first time, physical connectivity was established linking the abundant *Sargassum* populations in the GOM to nutrient-poor populations in the Sargasso Sea, helping to explain Ryther's Paradox[1].

Beginning in 2011, a new region of concentrated *Sargassum* biomass developed in the Tropical Atlantic Ocean south of the Sargasso Sea[12,13], where it had not been previously observed[3]. This new region may have been seeded by an extreme negative phase of the North Atlantic Oscillation in 2009 to 2010 that provided windage to transport *Sargassum* from the Sargasso Sea to the east and ultimately into the North Equatorial Current and central Tropical Atlantic Ocean[14], although this is not evident from satellite imagery. Long-term satellite data, numerical particle-tracking models, and field measurements indicate that a newly formed Great Atlantic *Sargassum* Belt (GASB) has recurred annually since 2011 and extended up to 8850 km from the west coast of Africa to the GOM, peaking in 2018[15]. Over its broad distribution, the GASB can be supported by N and P inputs from a variety of sources including discharges from the Congo, Amazon, and Mississippi rivers[5,15–17], upwelling off the coast of Africa[15,16], vertical mixing[5], equatorial upwelling[18], atmospheric deposition from Saharan dust, and biomass burning of vegetation in central and south Africa[16,19].

Similar to the recent development of macroalgal blooms in the Yellow Sea and the East China Sea[20,21], the increasing golden tides of *Sargassum* in the GOM and GASB could be ecological indicators of large-scale, oceanic eutrophication[15,22]. Excessive biomass strandings of *Sargassum* have had catastrophic consequences on ecosystem and human health in coastal areas, negatively impacting seagrasses[23], coral reefs[24,25], and a number of suitable sea turtle nesting and hatching areas[26]. *Sargassum* removal from Texas beaches during earlier, less severe inundations was estimated at $2.9 million per year[27] and Florida's Miami-Dade County alone estimated recent removal expenses of $45 million per year. The Caribbean-wide clean-up in 2018 cost $120 million, which does not include decreased revenues from lost tourism. *Sargassum* strandings also cause respiratory issues from the decaying process and other human health concerns, such as increased fecal bacteria. During large-scale strandings in 2018, more than 11,400 residents in Martinique and Guadeloupe were diagnosed with acute exposure to toxic $H_2S$ gas produced by decaying *Sargassum*[28].

Increases in harmful algal blooms (HABs) in recent decades are related to global increases in nutrient pollution[29,30]. Human activities have greatly altered global C, N, and P cycles, and N inputs are considered now high risk and above a safe planetary boundary[31]. Based on scientific research, population growth and land-use changes have increased N pollution and degradation of estuaries and coastal waters since at least the 1950s[30,32–34]. Despite decreases in N loading in some coastal watersheds, N:P ratios remain elevated in many rivers compared to historic values[35]. Although the relative importance of N vs. P limitation in the open oceans has been debated[36,37], previous analyses of tissue C:N:P data suggest that both N and P potentially limit the growth of pelagic *Sargassum* over its broad geographic range[5,6]. Here, the objective was to better understand the effects of N and P supply on *Sargassum*, where a unique baseline tissue C:N:P data set from the 1980s[5,6] are compared with more recent samples collected since 2010 (Fig. 1).

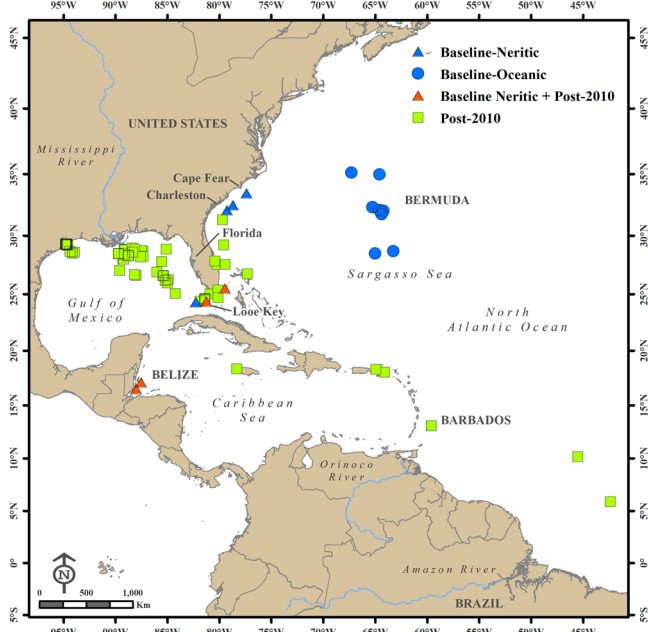

**Fig. 1 *Sargassum* collection locations.** Locations in the North Atlantic Ocean where *Sargassum* samples were collected during the 1980s baseline study[5] (blue), post-2010 collections (green), and during both time frames (orange).

## Results

A total of 488 tissue samples of *Sargassum spp.* were collected during various research projects and cruises in the NA basin between 1983 and 2019. The baseline 1980s samples (41) included

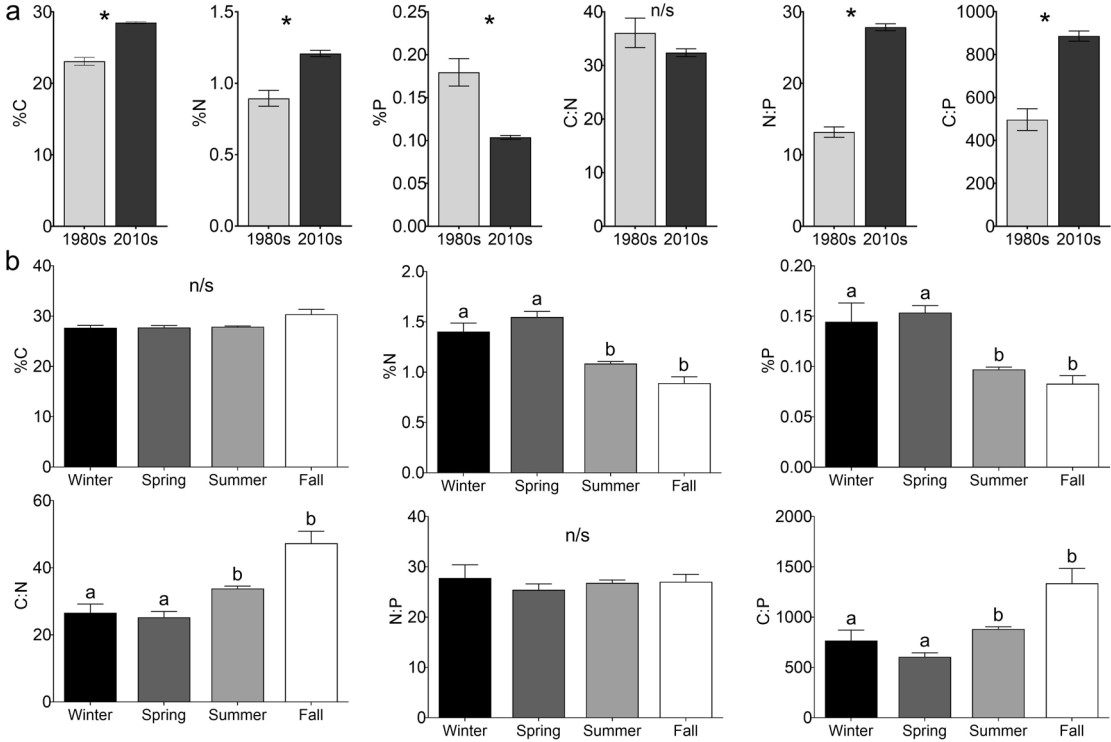

**Fig. 2 *Sargassum* tissue nutrient contents.** Tissue elemental composition and C:N:P stoichiometry (mean ± SE) of *Sargassum natans* and *S. fluitans* collected throughout the NA in the 1980s and post-2010. **a** by decade with asterisks representing significant differences and **b** by Northern Hemisphere meteorological season with different lowercase letters representing significant differences identified with Tukey HSD test; "n/s" denotes a non-significant (*P* > 0.05) ANOVA result.

seasonal sampling of *S. fluitans* (21 samples) and *S. natans* (20 samples) at offshore Looe Key reef in the lower Florida Keys in 1983 and 1984 and a broader geographic sampling in 1986 and 1987 from neritic stations offshore the Florida Keys (Looe Key, Dry Tortugas), Gulf Stream (Miami, FL; Charleston, SC; Cape Fear, NC), and Belize, Central America (Glovers Reef, Belize City). Oceanic stations included the northern, central, and southern Sargasso Sea (Fig. 1)[6]. Seasonally these baseline samples consisted of winter (2), spring (15), summer (20), and fall (4) collections. Since 2010, additional samples (447 total) of *S. fluitans* (302) and *S. natans* (145) were collected in a variety of locations in the wider NA, including Looe Key, western Florida Bay, the Gulf Stream, coastal waters along the east and west coasts of Florida, various stations in the GOM, Belize, the Caribbean region, and in the Amazon River plume (Fig. 1). The post-2010 samples also spanned winter (28) spring (97), summer (327), and fall (36).

**Changes in *Sargassum* tissue chemistry.** Tissue analysis of *Sargassum* over broad areas of the NA revealed significant changes in N and P contents since the 1980s, indicating widespread N enrichment and increased P limitation. %N and %C increased concurrent with a decrease in %P in *Sargassum* tissue from the 1980s to 2010s (Fig. 2a). Elemental composition varied significantly between these two decades (MANOVA, Pillai's lambda = 0.201, $F_{3,470}$ 39.4, *P* < 0.001; Supplementary Table 1). Subsequent univariate analyses revealed significant increases (23%) from the 1980s to the 2010s for %C (ANOVA, $F_1$ 53.8, *P* < 0.001) and %N (35%; ANOVA, $F_1$ 5.01, *P* = 0.026), while %P decreased significantly (−42%; ANOVA, $F_1$ 31.4, *P* < 0.001) over the long-term study (Fig. 2a). The C:N:P ratios also varied by decade (MANOVA, Pillai's lambda = 0.236, $F_{3,470}$ 48.4, *P* < 0.001; Supplementary Table 2). Notably, the biggest change was the N:P ratio, which increased significantly (111%; ANOVA, $F_1$ 93.4, *P* < 0.001). C:P ratios also increased similarly

(78%; ANOVA, $F_1$ 44.9, *P* < 0.001; Fig. 2a). Although the C:N ratio decreased (−10%) from the 1980s to the 2010s, this change was not significant (ANOVA, $F_1$ < 0.001, *P* = 0.956; Supplementary Table 2, and Fig. 2a). As such, the decadal patterns of increasing %C and %N with decreasing %P observed in *Sargassum* are consistent with observed changes in molar C:N:P ratios.

Seasonal patterns were also observed in elemental composition of *Sargassum* with higher %N and %P in the winter and spring (Fig. 2b). Elemental composition varied significantly with season (MANOVA, Pillai's lambda = 0.147, $F_{9,1416}$ 8.13, *P* < 0.001; Supplementary Table 1). Tissue %N (ANOVA, $F_3$ 17.1, *P* < 0.001) and %P (ANOVA, $F_3$ 16.7, *P* < 0.001) was significantly higher during winter and spring than in summer and fall, but %C was not seasonally variable (ANOVA, $F_3$ 2.58, *P* = 0.053). Further, tissue C:N:P ratios also varied with season (MANOVA, Pillai's lambda = 0.115, $F_{9,1416}$ 6.25, *P* < 0.001; Supplementary Table 2). Both the C:N (ANOVA, $F_3$ 13.4, *P* < 0.001) and C:P (ANOVA, $F_3$ 12.5, *P* < 0.001) ratios were significantly lower in the winter and spring compared to the summer and fall (Supplementary Table 2). The N:P ratio was not seasonally variable (ANOVA, $F_3$ 0.930, *P* = 0.427; Supplementary Table 2). These seasonal patterns demonstrate higher %N and %P contents in the winter and spring with no seasonal fluctuations in %C or N:P ratio (Fig. 2b).

Significant interactions between season and decade were observed in *Sargassum* tissue chemistry. For elemental composition, this interaction was significant (MANOVA, Pillai's lambda = 0.055, $F_{9,1416}$ 2.95, *P* = 0.002; Supplementary Table 1, Supplementary Fig. 1). For %C the interaction of season and decade was not significant (ANOVA, $F_3$ 1.90, *P* = 0.129) but there were significant interactions for %N (ANOVA, $F_3$ 3.00, *P* = 0.030) and %P (ANOVA, $F_3$ 3.91, *P* = 0.009). For %N, all seasons increased from the 1980s to 2010s, except for winter, which slightly

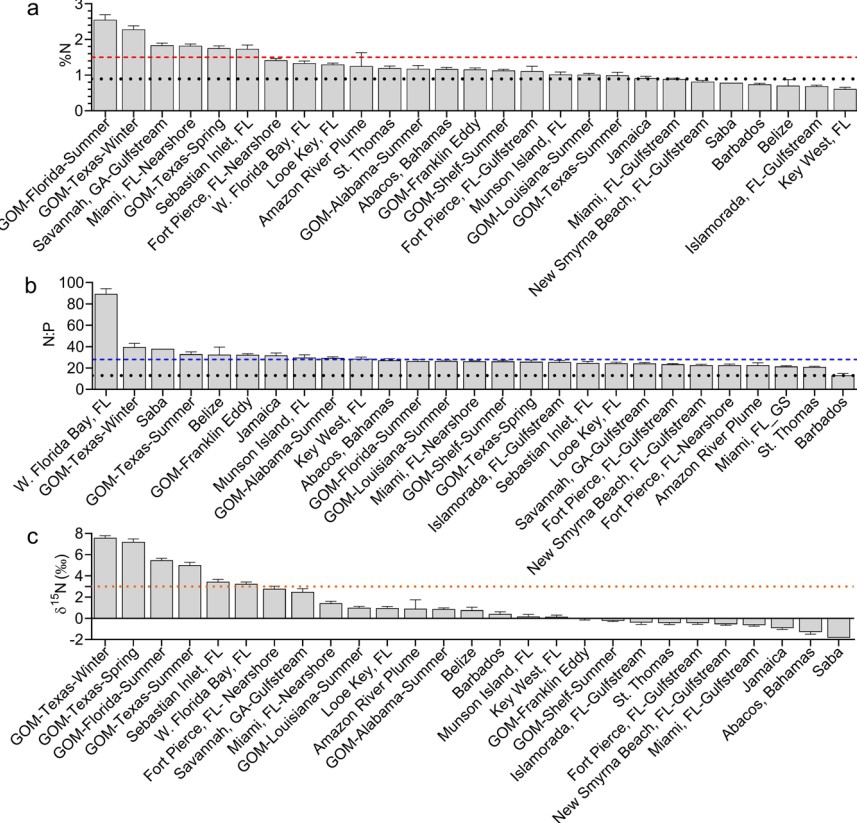

**Fig. 3 Post-2010 *Sargassum* tissue nutrient contents by location.** Post-2010 *Sargassum* tissue nutrient contents by location (mean ± SE), as well as Northern Hemisphere meteorological season for Gulf of Mexico (GOM) samples, indicating where %N and N:P ratios were greater than the 1980s baseline mean for the entire dataset (black dotted lines). **a** For %N, values have significantly increased from the 1980s (decadal mean = 0.89%) to post-2010 (decadal mean = 1.21%); %N values >1.5 (red dashed line) are considered non-limiting to macroalgal growth[43]. **b** N:P ratios have significantly (111%; ANOVA, $F_{=1}$ 93.4, $P < 0.001$) increased from the 1980s (decadal mean = 13.2) to post-2010 (decadal mean = 27.8, blue dashed line). **c** Enriched $\delta^{15}N$ values (>+3‰, orange dotted line) are indicative of urbanized wastewater discharges, while more depleted values are indicative of $N_2$ fixation, atmospheric deposition, and upwelling.

decreased (Supplementary Fig. 1). Conversely, %P decreased between decades for all seasons, except winter, which slightly increased (Supplementary Fig. 1). These interactions suggest that 1980s winter had a different pattern than the other seasons, which may be an artifact of the very small sample size (2) from just one location (Looe Key) for 1980s winter.

**Geographic patterns in *Sargassum* tissue chemistry.** Overall, the %N of *Sargassum* spp. increased 35% over the period of study, while %P decreased by 44%, resulting in more than a doubling of the N:P ratio from 13:1 to 28:1, well above the Redfield Ratio of 16:1. The highest %N, N:P ratios, and stable nitrogen isotope values ($\delta^{15}N$) were in neritic waters heavily influenced by river discharges and land-based runoff (Fig. 3, Supplementary Fig. 2, and Supplementary Table 3). The overall range of %N was 0.15 to 3.05% with the highest mean %N observed in coastal waters of the GOM (2.55% in summer offshore of Florida's west coast, 2.28% in winter offshore of Texas), Florida's east coast (1.82% offshore of Miami, 1.73% near Sebastian Inlet, 1.33% in western Florida Bay), southeast United States (1.84% offshore Savannah, GA) and the offshore Amazon plume (1.25%; Fig. 3a and Supplementary Fig. 2a). The lowest %N was observed offshore of Key West, FL (0.61%; Fig. 3a). Twenty of the post-2010 sampling events had % N values greater than the mean %N from the 1980s (Fig. 3a). The overall range for N:P ratios was 4.66 to 99.2 with the highest in western Florida Bay, FL (89.4), followed by locations in the GOM and Caribbean (Fig. 3b and Supplementary Fig. 2b). The lowest

N:P ratios were observed in the eastern Caribbean at St. Thomas (20.9) and Barbados (13.0; Fig. 3b). Twenty-six of the post-2010 sampling events had N:P ratios greater than the mean N:P from the 1980s (Fig. 3b). $\delta^{15}N$ values were variable with an overall range of −5.58 to +8.99‰, indicating multiple sources of N were available to *Sargassum* (Supplementary Fig. 2c). High values (>+5‰) occurred along the urbanized Texas coast that is also affected by the Mississippi River plume (Supplementary Fig. 2c). The lowest $\delta^{15}N$ values occurred at Saba (−1.83‰) in the Leeward Islands of the northeastern Caribbean (Fig. 3c). $\delta^{15}N$ values of *Sargassum* collected in the Gulf Stream were also generally low (<−1‰), except offshore of Savannah, GA (+2.5‰; Fig. 3c).

## Discussion

In a series of shipboard experiments during the 1980s, *Sargassum* productivity and growth was enhanced by enrichment with both nitrate[5] ($NO_3^-$) and soluble reactive phosphorus[38] (SRP), which resulted in higher tissue levels of N and P. In oligotrophic surface waters of the NA, dissolved inorganic N (DIN) and SRP concentrations are higher within *Sargassum* windrows compared to adjacent waters[5,39]. This localized enrichment has allowed *Sargassum* to exploit ammonium-rich excretions from associated fishes and invertebrates[6], recycled nutrients from microbial mineralization of particulate organic matter[40] (POM) and dissolved organic N forms such as urea and amino acids[41] through its long evolutionary history. While several forms of dissolved

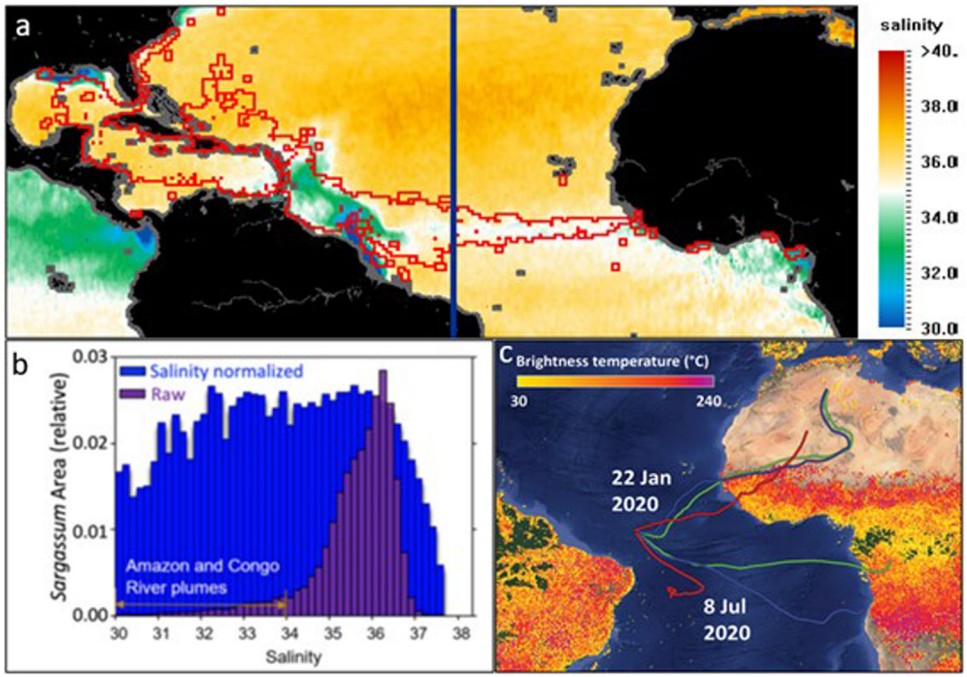

**Fig. 4 Spatial distribution of *Sargassum* in relation to salinity and aerosol trajectories. a** *Sargassum* distribution (red empty squares) overlaid on salinity derived from the Soil Moisture Ocean Salinity (SMOS) satellite mission, both for July 2018. The blue line marks the longitude of 38°W that the Amazon River plume hardly reaches. **b** Distribution of *Sargassum* as a function of salinity (purple) for 2011 to 2019, where statistics are calculated from 108 monthly mean maps over the cumulative *Sargassum* footprint. Here, the raw data (purple) shows *Sargassum* areal coverage in each salinity increment, relative to the total coverage (i.e., all purple bars sum to 1.0). The *Sargassum* coverage in each salinity increment is divided by the water area in the same salinity increment, resulting in the "salinity normalized" distribution (blue). The *Sargassum* distribution data were obtained on National Centers for Environmental Information (NCEI) Accession 0190272[14]. **c** Hybrid Single-Particle Lagrangian Integrated Trajectory model (HYSPLIT) air mass back trajectories (10-day) from 3°N, 34.6°W at 500 m (red), 1000 m (blue), and 1500 m (green), shown with the active fires in Africa and South America observed during Apr 2020 to Mar 2021 (from firms.modaps.eosdis.nasa.gov/download[85]).

nitrogen are available to *Sargassum*, ammonium ($NH_4^+$) uptake is most efficient[42].

More recently, the significant increase in tissue N (+35%) and upward shifts in N:P ratios (+111%) since the 1980s suggests that *Sargassum* is now exploiting the global trend in N enrichment. In the 1980s %N of *Sargassum* averaged 0.89% compared to higher, non-limiting values for macroalgae (>1.5%[43]) observed recently in the GOM, peninsular Florida, and the Amazon Plume (Fig. 3). Because of anthropogenic emissions of oxides of N ($NO_x$), the $NO_x$ deposition rate is about fivefold greater than that of pre-industrial times largely due to energy production and biomass burning[44]. Production of synthetic fertilizer N has increased ninefold, while that of P has increased threefold since the 1980s[30] contributing to a global increase in N:P ratios. Notably, 85% of all synthetic N fertilizers have been created since 1985[45], which was shortly after the baseline *Sargassum* sampling began at Looe Key in 1983. The quantity of global N fixation for fertilizer production and P flowing into the oceans was estimated at 121 and 9.5 million tons/yr, respectively[46], yielding an anthropogenic N:P molar ratio of 28:1, identical to the mean N:P molar ratio of 28:1 measured in *Sargassum* since 2010.

A strong connection of *Sargassum* areal cover to land-based runoff is evidenced by the highest tissue %N values occurring in areas influenced by reduced salinity from river discharges and terrestrial runoff (Figs. 3a and 4a). Statistical analysis of the *Sargassum* cover in different salinity ranges from 2011 to 2019 shows that the bulk of *Sargassum* biomass occurs at oceanic salinities of ~36 (Fig. 4b). However, when *Sargassum* cover (or biomass) is normalized by water area in each salinity band (blue bars in Fig. 4b), the distribution is rather flat across the salinity range of 32 to 36 with 32.4 to 33.5 containing slightly higher

abundance of *Sargassum* than 33.5 to 35, indicative of riverine influence. For waters with salinity <31, *Sargassum* abundance is lower, possibility due to the lower growth rate at low-salinity waters[47]. The Mississippi River[48,49] and South Florida's Everglades and coastal urban belt[50,51] have experienced trends of increasing N flux and increasing N:P ratios[35]. For the GOM, the combined annual mean streamflow for the Mississippi and Atchafalaya rivers represents about 80% of the freshwater discharge to the GOM and accounts for 90% of total N load and 87% of the total P load discharged annually to the GOM[52]. Increasing nitrogen (mostly $NO_3^-$) along with other nutrients are a cause of hypoxia in a large dead zone along the Louisiana-Texas coast[53,54], where the highest *Sargassum* tissue %N values (Fig. 3a) were observed in waters that tend to have lower salinity (Fig. 4a). In addition, the N:P ratio of the Mississippi River and northern GOM increased from 9 to 15 and 16 to 24, respectively, between 1960s and the 1980s[54], indicating that this stoichiometric shift began prior to the current study. A 30-year study between 1984 and 2014 at Looe Key reef showed over a twofold increase in seawater DIN and the DIN:SRP ratio, and threefold increase in tissue N:P ratio in a variety of reef macroalgae[50]. Some of the *Sargassum* collections in the present study were made in blue water offshore of Looe Key reef and paralleled this pattern of tissue N enrichment and an increased N:P ratio from 11.2 in the 1980s to 24.2 since 2010.

The Amazon River is the largest river in the world and accounts for 20% of the world's total river discharges. Data from the 13 Carbon in the Amazon River Experiment (CAMREX) cruise surveys of the Amazon River between 1982 and 1991 show strong, statistically significant correlations between $NO_3^-$ flux and discharge and between SRP flux and discharge (Fig. 5a). The

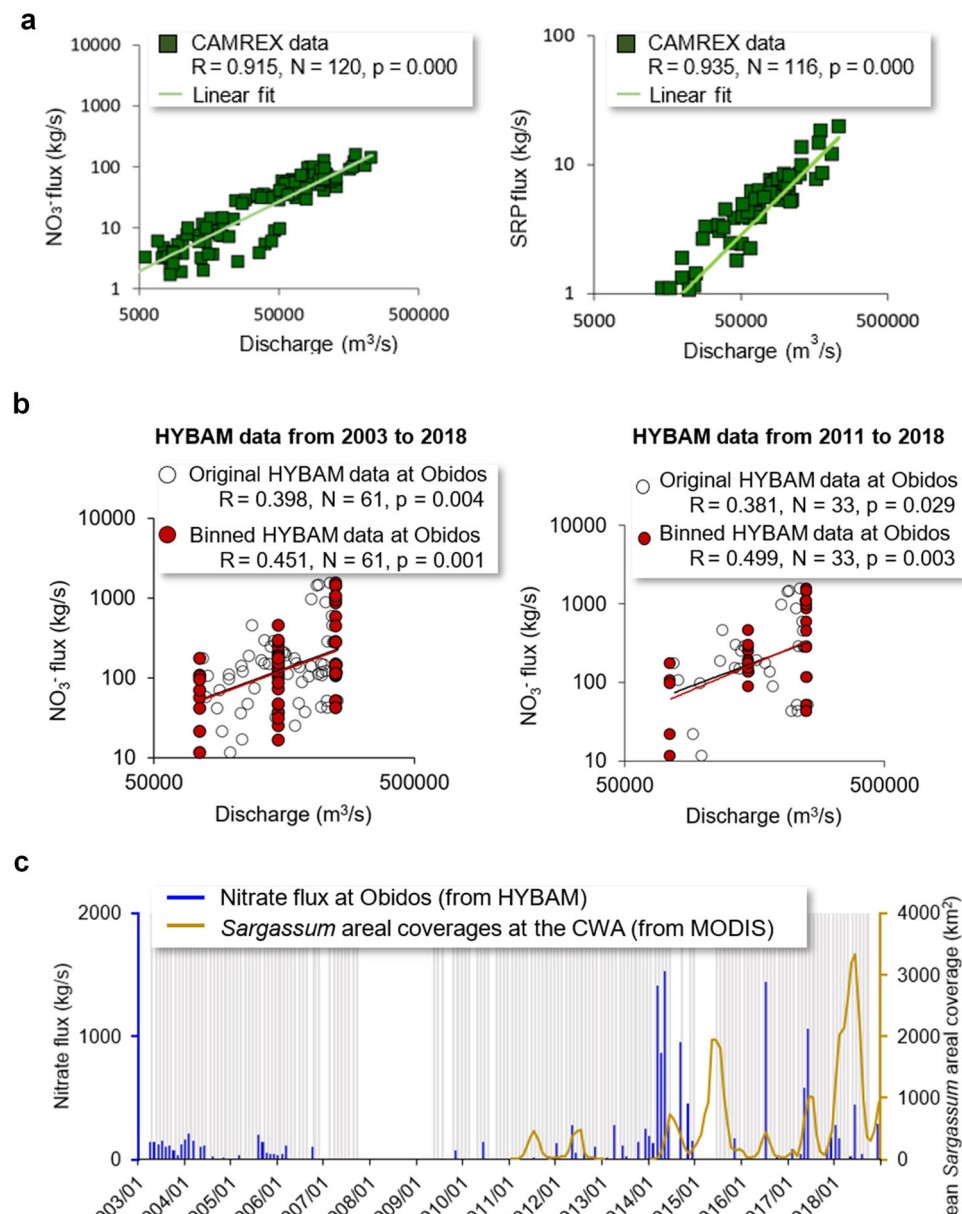

**Fig. 5 Nutrient flux from the Amazon River and long-term *Sargassum* trend. a** Nitrate (NO₃⁻) and soluble reactive phosphorus (SRP) flux from all stations of the 13 CAMREX cruises[83] (1982 to 1991) are highly correlated with river discharge. These stations are within a 2000 km reach of the Brazilian Amazon River mainstream. Solid lines mark the power law regression lines. **b** NO₃⁻ flux at Obidos from the HYBAM database[100] is also correlated with discharge, and the correlation is higher if data are binned to different discharge groups. **c** NO₃⁻ flux at Obidos from the same HYBAM dataset shows apparent increases in recent years. The recent mean monthly *Sargassum* areal coverages obtained from Wang et al. (2019) in the Central West Atlantic (CWA, 0ºN to 22ºN, 63ºW to 38ºW) are also shown as reference. All months before 2011 show *Sargassum* coverages. The shaded gray bars indicate when quality controlled HYBAM nutrient flux measurements are available, but data in some months are too low (<0.1 mg/L) to be visible due to the scale.

monthly data at the Obidos station of the Amazon River from the Hydrology and Geochemistry of the Amazon basin (HYBAM) observatory also show a positive, statistically significant ($P =$ 0.049) correlation between NO₃⁻ flux and discharge (Fig. 5b). There have been recent increases in NO₃⁻ fluxes at the Obidos station (Fig. 5b), especially between 2014 and 2016, with lower fluxes between 2016 and 2018 (Fig. 5c). Although a direct annual correspondence between NO₃⁻ flux and *Sargassum* amount is not apparent, the general trend of increased NO₃⁻ and SRP after 2014 suggests that river discharge may have supported *Sargassum* growth around the plume in subsequent years, especially 2015, 2017, and 2018. The HYBAM Obidos results are consistent with the limited field measurements of NO₃⁻ and SRP in the offshore

river plume (salinity 16 to 34) in 2010 and 2018[15,55], especially when compared with the lower historical nutrient values in the 1960s[56] and early 2000s[57]. Such increases in nutrient fluxes could be partially due to Amazonian deforestation (+25% since 2010)[15], which has been shown to alter the hydrochemical balance of streams and soil chemistry. In addition, extreme flooding events in the Amazon basin in this time frame[58] combined with increased fertilizer use (+67% since 2010)[15] could also contribute. Although the Amazon River plume hardly reaches waters east of 38°W and therefore its contributions to *Sargassum* blooms in the eastern Tropical Atlantic can be neglected, its nutrient flux may have fueled recent blooms since 2014 in the central West Atlantic (Fig. 5c). The importance of the Amazon River nutrient flux to

*Sargassum* growth was also apparent in *S. natans* collected in plume waters (~10°N, 45.5°W; salinity ~33.8) in late August 2019 that averaged 2.4%N, a very high value that was exceeded only by *S. fluitans* collected in the Mississippi River plume off the Texas and west Florida coasts. Interestingly, while the overall effect of species was not significant (see Supplement), at this location *S. fluitans* was much less enriched in %N (mean = 0.66%) than *S. natans*. Considering that the Amazon Basin dominates P flux to the NA[59] and that land-based P exports can reach the open ocean[60] the productivity of *Sargassum* could be enhanced in this region of the western Tropical Atlantic by increased P availability. This is supported by the lowest N:P ratios observed in 2015 in Barbados in the present study (Fig. 3b), which is directly influenced by the Amazon River plume.

Seasonal changes in C:N:P contents of *Sargassum* closely matched patterns in nutrient flux from the Mississippi and Amazon rivers, further suggesting these river discharges support seasonal growth patterns. These river discharges increase from winter through spring and peak in early summer[15,35,49] and could support the lower C:N and C:P ratios in *Sargassum* during this period; in contrast, the higher C:N and C:P ratios in summer and fall indicate nutrient limitation resulting from reduced river discharges[5,6]. These data suggest seasonal nutrient control of bloom formation by seasonal river discharges, particularly in the GOM, where *Sargassum* cover expands in the spring and peaks in summer months (Supplementary Fig. 3)[5,15,61]. Its air bladders allow *Sargassum* to float and form dense mats which are advected by ocean currents thus allowing episodic access to buoyant, lower salinity, nutrient-enriched river plumes, which can extend for thousands of kilometers from shore[15] (Fig. 4a). Similar to the temperate kelps, *Laminaria longicruris*[62] (synonym *Saccharina latissimi*) and *Macrocystis pyrifera*[63], the floating tropical *Sargassum* spp. appear to be responders rather than anticipators among macroalgae[64] by sequestering seasonally available N and P to support annual growth patterns. A seasonal NO$_3^-$ related growth strategy occurs in the temperate kelp *L. longicruris*, which assimilates and stores NO$_3^-$ in winter months when its available to support maximum growth rates in the spring and into July, after which tissue N and growth rate decline[62].

The wide range of δ$^{15}$N values in *Sargassum* tissue from −2 to +8‰ reinforces previous suggestions that a variety of N sources support growth of *Sargassum* over its broad geographic range[5,15,65]. The δ$^{15}$N values of wet atmospheric deposition across the United States are relatively low, ranging from −11 to +3.5‰ with a median value of −3.1‰ (n = 883)[66] and are within the low end of the range of δ$^{15}$N in *Sargassum*. Similarly, in Bermuda, rainwater NH$_4^+$ δ$^{15}$N values ranged from −12.5 to +0.7‰[67]. Synthetic fertilizer N, which has increased ninefold since the 1980s, has δ$^{15}$N values ranging from −2 to +2‰[68] and is the mid-range for most *Sargassum* values in this study. More enriched values of +2.5 to +4.8‰ are indicative of upwelled NO$_3^-$ in the upper 200 m of the NA[69,70]. Higher values (+3 to +20‰) are indicative of urban wastewater from terrestrial runoff where fractionation associated with volatilization of NH$_4^+$ and denitrification of NO$_3^-$ occur[71]. This enrichment of *Sargassum* tissue is evident in the highest δ$^{15}$N values ranging between +3 to +8‰ along urbanized coastal waters in Texas and Florida, illustrating the effect of anthropogenic nitrogen enrichment. The mean δ$^{15}$N value of POM in the Mississippi River is ~+7‰[72] and δ$^{15}$N enrichment of *Sargassum* by +2‰ has been reported for neritic compared to oceanic regions in the GOM[65], as well as macroalgal blooms on coral reefs downstream of sewage outfalls in South Florida (+6 to +8‰)[51]. In the Amazon River floodplain, δ$^{15}$N of phytoplankton and macrophytes range from +4.7 to +5.5‰ respectively[73], indicating that the plume could contribute to δ$^{15}$N enrichment of *Sargassum*. These findings suggest

that episodic N enrichment in highly populated tourist areas of the Caribbean could help sustain *Sargassum* growth and bloom continuation.

Natural N and P sources, such as upwelling and N$_2$ fixation, could further support *Sargassum* growth and would be especially important in offshore and oceanic locations of the GASB[13]. Upwelling occurs at the shelf break in the southeastern United States[74,75] and in the eastern equatorial Atlantic[76] and could supply NO$_3^-$ to *Sargassum*. N$_2$ fixation by the cyanobacterial epiphyte *Dichothrix fucicola* occurs in *Sargassum* windrows and can provide from 2 to 32% of the N needs[77,78], which would result in δ$^{15}$N values close to 0‰. N$_2$ fixation plays a prominent role in N-nutrient cycling in the Amazon and Congo river plumes[57,79]. N$_2$ fixation by diatom diazotroph associations (DDAs) and *Trichodesmium* support 11% of total primary production in the mesohaline section of the Amazon River plume (salinity ~32 to 33)[57]. During a study in which the plume extended into the Caribbean Sea, diazotrophy by DDAs supplied ~25% of water column N demand[80]. In the eastern equatorial Atlantic (Gulf of Guinea), N$_2$ fixation rates were 2 to 7 times higher when upwelling occurred as compared to non-upwelling conditions, as a result of low NO$_3$:SRP ratios in upwelled waters that leave excess SRP that stimulates N$_2$ fixation[76]. Considering the high N:P ratio of *Sargassum* that now occurs in the NA basin, such excess SRP in the upwelled water could stimulate growth of *Sargassum* in the eastern Tropical Atlantic.

Atmospheric deposition (dry and wet) of lithogenic and anthropogenic-sourced aerosols can supply the central Atlantic with major and trace nutrients that could further support *Sargassum* growth. Aerosol back trajectories (examples shown in Fig. 4c) show that winds over the central Atlantic change seasonally but are predominantly from northern Africa, where Saharan dust originates, and central and southern Africa, where biomass burning generates anthropogenic-type aerosols[19,81]. The largest atmospheric supply of nutrients, like Fe and P, comes from seasonal Saharan dust plumes, but the low solubility of these elements in mineral dust limits their bioavailability[82–84]. In contrast, biomass burning in central and southern Africa (as shown as active fires in Fig. 4c[85–87]) can deliver nutrients like N, P, and Fe to the central Atlantic. Like wind patterns, biomass burning also varies seasonally with the Northern (Southern) Hemisphere burn season occurring in November to March (May to October)[88]. While nutrient concentrations in aerosols produced during biomass burning might be lower than in Saharan dust, the nutrient solubilities are higher, potentially providing a source of more bioavailable nutrients to the Atlantic surface ocean[83,89–91]. Fluxes of aerosols from both dust and biomass burning appear to have increased over the past 150 years, according to models of past, present, and future changes to the atmospheric deposition in the North and Central Atlantic[92]. While the fluxes of dissolved P to the NA have likely increased since 1850, which would alleviate P limitation, the fluxes of dissolved Fe have increased faster resulting in higher dissolved Fe/dissolved P ratios[91,92], which could enhance N$_2$ fixation and balance increased inputs of bioavailable P. Future atmospheric inputs are difficult to predict as aerosol production and processing rely heavily on economic and human factors, including restrictions on biomass burning, fossil fuel consumption, and industrial pollution. Nonetheless, seasonal inputs of natural and anthropogenic-driven aerosol nutrients could at least partially alleviate P and/or Fe limitation, resulting in increased *Sargassum* growth and abundance across the central Atlantic Ocean.

Almost 50 years ago, scientists recognized that nutrient addition through use of fertilizers can destabilize food webs, leading to loss of biodiversity and ecosystem function through the so called paradox of enrichment[93,94]. During that time, global river discharges showed a trend of decreasing N:P due to human activities

and P was considered the primary limiting nutrient in surface waters[95]. Concurrently, the precept that N, rather than P, was driving marine eutrophication was introduced to the scientific community[33]. Since then, N, Fe, and silica have been widely considered to be the most important nutrients that limit phytoplankton growth in the oceans, although mounting evidence is supporting an emerging paradigm in oceanography that P plays a primary role in the Atlantic basin[96–98]. Recent reviews now show that N:P ratios of rivers are increasing, despite attempts to mitigate application of N fertilizers[35]. The empirical data presented here for *Sargassum* supports not only a primary role for P limitation of productivity, but also suggests that the role of P as a limiting nutrient is being strengthened by the relatively large increases in anthropogenic N supply from terrestrial runoff, atmospheric inputs, and possibly other natural sources such as $N_2$ fixation[96]. The increased P limitation in *Sargassum* could be compensated for by its relatively high capacity for alkaline phosphatase activity, which allows it to sequester SRP from dissolved organic P compounds, a physiological characteristic of adaptive value to growth in oligotrophic waters[5]. Considering the negative effects that the GASB is having on the coastal communities of Africa, the Caribbean, GOM, and South Florida, more research is urgently needed to better inform societal decision-making regarding mitigation and adaptation of the various terrestrial, oceanic, and atmospheric drivers of the *Sargassum* blooms.

## Methods

**Sample collection.** *Sargassum* samples in the 1980s were collected mostly from University-National Oceanographic Laboratory System research vessels, including the R/V *Columbus Iselin* (Loop Current, Gulf Stream, Sargasso Sea), R/V *Calanus* (Belize), RV *Cape Hatteras* (Sargasso Sea, Gulf Stream, Belize), and R/V *Weatherbird* (Sargasso Sea); for blue waters offshore Looe Key in the lower Florida Keys, *Sargassum* was collected from a small boat (20′ Mako). Since 2010, *Sargassum* has been collected from the R/V *Point Sur* (GOM) and the R/V *Thomas G. Thompson* (Amazon plume). Other samples were collected by volunteers on private vessels and the M/V *Ocearch* (Gulfstream). Windrows of *Sargassum* spp., which result from Langmuir circulation that aligns *Sargassum* parallel with the wind direction, were frequently encountered at various locations during the research cruises. For all sampling events, *Sargassum* spp. were collected from small boats either by divers or with a dip net and sorted into the species and morphotypes, *S. natans* I and *S. fluitans* III per Parr (1939)[4]. After collection, the plants were placed in clean plastic bags in a cooler. Upon return to the lab or research vessel, the samples were separated into replicate (*n* = 2 to 3/species for each location and sampling) composite samples (6 to 10 thalli/species), rinsed briefly (3 to 5 s) in deionized water, cleaned of macroscopic epizoa and epiphytes, dried in a laboratory oven at 65 to 70 °C for 48 h, and powdered with a mortar and pestle[5].

For both the 1980s and 2010s tissue analysis, total C and N were determined on a Carlo-Erba CHN Combustion Analyzer, while total P was determined by persulfate digestion followed by analysis for SRP using either a Bausch and Lomb Spectronic 88 or an Alpkem 300 series autoanalyzer. The resulting tissue %C, %N, and %P data were used to calculate molar C:N:P ratios. Additional analysis of 2010 tissue (427) $\delta^{15}N$ was conducted on a Thermo Delta V IRMS coupled to a Carlo Erba NA1500 CHN-Combustion Analyzer via a Thermo Conflo III Interface.

**Statistical analysis.** The relationship between *S. fluitans* and *S. natans* elemental composition (%C, %N, %P) and molar ratios (C:N:P) with species, decade, and season were analyzed using multivariate and subsequent univariate general linear models (MANOVA and ANOVA) in Minitab 19 Statistical Software. All variables were non-normal therefore log transformation was attempted prior to analyses and model fit was assessed through examination of residuals. While log transformation improved the normality, shape, and residual distribution of %N, %P, C:N, N:P, and C:P, %C was not improved and thus the raw values for this parameter were used in analyses. Significant univariate factors and interactions were assessed with Tukey's pairwise comparisons. To better understand nitrogen sources supporting *Sargassum* bloom growth and development, %N, N:P ratios, and $\delta^{15}N$ of post-2010 samples were compared by location with ANOVA using similar methods as above in Minitab 19 Statistical Software. Statistical significance was considered at $P < 0.05$ for all analyses.

**Satellite-measured distributions of *Sargassum* and salinity and field-measured river nutrient concentrations.** Pelagic *Sargassum* distributions covering the GOM and central Atlantic Ocean were derived from MODIS measurements

and the data were acquired from NCEI Accession 0190272([15]). Surface salinity distributions were obtained from SMOS Earth Explorer mission and the data were accessed on https://www.catds.fr/Products/Available-products-from-CPDC). Amazon River discharge and water chemistry data, including $NO_3^−$, and $PO_4^{3−}$ concentrations, measured at the Obidos station during 2003 to 2018 were downloaded from HYBAM database (http://www.ore-hybam.org/index.php/eng/Data). Similar parameters were also from 1981 to 1991 collected during cruises for the CAMREX[99] were also analyzed.

## Data availability

All data used in this study are available in the main text or the supplemental materials. The raw data that support the findings of this study are available from the corresponding author upon reasonable request. All *Sargassum*-relevant imagery data products are available through the *Sargassum* Watch System (SaWS, https://optics.marine.usf.edu/projects/saws.html).

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

## Acknowledgements

We thank J. Bishop, M. Clark, W. Matzie, M. Littler, D. Littler, R. Brown, B. Brown, A. Tewfik, D. English, J. Cannizzaro, L. Wilking, A. Feibel, J. Franks, J. Nomura, E. Cheung, S. Brugger, D. Baladi, D. Milmore, and J. Conover for help in collecting and processing *Sargassum* samples. E. Carpenter and D. Anderson provided helpful comments on the research. This work was funded by the US NASA Ocean Biology and Biogeochemistry Program (80NSSC20M0264, NNX16AR74G) and Ecological Forecast Program (NNX17AF57G), NOAA RESTORE Science Program (NA17NOS4510099), National Science Foundation (NSF-OCE 85–15492 and OCE 88–12055), "Save Our Seas" Specialty License Plate funds, granted through the Harbor Branch Oceanographic Institute Foundation, Ft. Pierce, FL, and a Red Wright Fellowship from the Bermuda Biological Station. A portion of this work was performed at the National High Magnetic Field Laboratory, which is supported by National Science Foundation Cooperative Agreement No. DMR-1644779 and the State of Florida. D.J.M. gratefully acknowledges the Holger W. Jannasch and Columbus O'Donnell Iselin Shared Chairs for Excellence in Oceanography, as well as support from the Mill Reef Fund. Special thanks to the captains and crews of the R/V *Columbus Iselin*, R/V *Calanus*, R/V *Cape Hatteras*, R/V *Weatherbird*, M/V *Ocearch*, R/V *Point Sur*, and R/V *Thomas G. Thompson*, especially technicians Jennifer Nomura, Emily Cheung, and Sonia Brugger, who collected the samples in the Amazon River plume in 2019. We acknowledge the use of data and/or imagery from NASA's Fire Information for Resource Management System (FIRMS) (https://earthdata. nasa.gov/firms), part of NASA's Earth Observing System Data and Information System (EOSDIS). Special thanks to Alexandra Music (FSU) for generating the map of active fires and HYSPLIT trajectories shown in Fig. 4c. This is contribution #2290 of the Harbor Branch Oceanographic Institute at Florida Atlantic University

## Author contributions

B.E.L. was responsible for collection and analysis of the *Sargassum* tissue, completed first draft of this manuscript, led data interpretation, and conceived the project. R.A.B. contributed to data analysis and interpretation, geospatial analyses, and manuscript editing. L.W.H. contributed to geospatial analyses. M.W contributed to *Sargassum* collections, data acquisition, and analyses. C.H. contributed to project conception, data acquisition, analyses, and interpretation. D.J.M. contributed data acquisition, analyses, and interpretation. S.L. contributed data acquisition, analyses, and interpretation. F.J.H. contributed data acquisition, analyses, and interpretation. P.L.M. contributed data acquisition, analyses, and interpretation.

## Competing interests

The authors declare no competing interests.

## Additional information

**Peer review information** *Nature Communications* thanks Pamela Fernandez, Candace Oviatt, and Mirta Teichberg for their contribtiuons to the peer review of this work. Peer review reports are available.

