## [Peer Review File · Nature Communications]

REVIEWER COMMENTS

Reviewer #1 (Remarks to the Author):

The authors have conducted an interesting study determining the effects of inorganic nitrogen availability on *Sargassum* sp. blooms in waters of the North Atlantic Ocean. They have found that the occurrence of these blooms are strictly associated with changes in inorganic nitrogen and phosphorus inputs from the last decades (terrestrial runoff and river discharges). The manuscript is very well written and easy to follow, but there a few points that I would like to pointed out.

Inorganic nitrogen and phosphorus play important roles in macroalgae physiology and ecology, limiting their growth. The current study have shown a similar pattern in the N:P ratio from river discharges and the N:P ratio in algal tissue, showing that this ratio have significantly increased in the past decades likely due to eutrophication. *Sargassum* sp. usually grows in oligotrophic waters, and hence it is particularly interesting evaluate the impact of eutrophication on their abundances and distribution patterns as their blooms have caused great ecological and social impacts in the North Atlantic ocean. Overall, I think this is a very nice study, and it will be of great importance for decision-making authorities.

One of my suggestions would be to include more details about the collection sites in the introduction and/or map (Fig.1) because the method section is at the end of the ms, and sometimes it is difficult to follow the result section. In addition, I saw in the method section that the authors have also measured $\delta^{13}\text{C}$ in the samples, why these results were not include or discussed through the ms? Are they relevant for the study or not?

On the discussion line 234 the authors mentioned that previous studies have measured N and P content in the same species. How these values are compare to those obtained in the current study? I could see the raw data in the Supplementary table S4, and they are quite low, except by the summer season where they are influenced by the river discharges. So, It would be nice to see a short paragraph describing or comparing these results with previous studies. This species was usually exposed to oligotrophic waters, and now it exposed to very euthropied waters, so it will be nice to compare the N and P content to see if they are capable to store it. Do you think they have change their growth strategy? Why limiting phosphorus does not limit their growth?

Line 288: delete soluble reactive phosphorus (described above).

Line 322 : how could you describe the growth pattern in this species, anticipator or responder? Can its growth strategies change across their distribution range?

Lines 321-325: how these attributes contribute to their nitrogen physiology (or nitrogen uptake strategies)? It is not clear how its air bladders or reproduction patterns can favour their distribution in nutrient-enriched waters.

Reviewer #2 (Remarks to the Author):

Nature

Communications

MS

NCOMMS--20--42996

Nutrient content and stoichiometry of pelagic *Sargassum* reflects increasing nitrogen availability in the Atlantic Basin

By B. E. Lapointe, R. A. Brewton, L. W. Herren, M. Wang, C. Hu, D. J. McGillicuddy, Jr., S. Lindell, F. J. Hernandez, P. L. Morton

General

Lapoint et al. have a priceless data set from 40 years ago to compare to a current nutrient content, nutrient type and isotope signature data set on pelagic *Sargassum* in the Gulf of Mexico, Florida Keys and open Atlantic waters. They make a convincing argument that N input through Mississippi River discharge and land sources to the GOM has

significantly increased with global growth of artificial fertilizers and contributed to increased N content of Sargassum there. They do not show a quantitative increase in the biomass of Sargassum in the GOM and that would be a useful addition to the argument in this paper. Their speculation that this increase in fertilization is also happening in the GASB is in line with the speculation of many other authors. However, the data base provided for this speculation is much smaller, the values much lower and the story weaker than the one for the GOM. The authors are correct to indicate more research is needed. This paper with improvements is a useful contribution to the ongoing puzzle of why Sargassum has increased so much in the GASB but does not yet solve the puzzle.

Specific

P1 line 40 might be first identified as "Ryther's" paradox.

P6 Results No actual results on a quantitative increase in Sargassum in the GOM and export to the NA is presented. It would be a useful contribution to include such data analogous to Figure 6d.

Figure abc Useful to indicate statistical significance of differences along the gradient.

Figure 4 b Useful to add line of global N:P ratio of 28 onto graph?

Figure 5a What is SMOS? What is the source of 2018 information on the distribution of Sargassum?

Figure 5b Salinity normalized to what? What is the raw data abundance and distribution for what year(s)? P15 line 257-259 explain calculation and abundance of Sargassum for what year(s).

Figure 5c What do the colors over land indicate -average area burned. Perhaps we need a key?

Figure 6abc Need reference for CAMREX cruises. It would be nice to show regressions for post 2010 data on nutrient discharge?

Figure 6d Need reference for HYBAM data set. Looks like max value for nitrate Y-axis could be 2000 kg/s. Why is there so little data for most years compared to 2003-4?

Provide reference for the source of the Sargassum data for the CWA. Data for 2018 shown for Figure 5 not included here?

P17 line 299 Location of 38°W should be noted on Figure 5a. Equatorial counter current in early summer may carry the Amazon River plume further offshore and to Africa? See distribution of lower salinities on Figure 5a.

P20 Sources and signatures of N-15 should usefully include Sargasso Sea, sewage, CWA, Congo and Amazon River discharge data.

P20 line 354-364 Do you suggest that N₂ fixation has increased in GASB in correlation with the bloom increases of Sargassum? If so it would be useful to provide further evidence of the reason for such an increase in N₂ fixation such as greater Fe input to the area.

P21 line 368-382 If there is any evidence that this aerosol source of nutrients has increased in correlation with the bloom increases of Sargassum in the GASB it would be useful to include it here.

P24 line 450 define NCEI.

Candace Oviatt

Reviewer #3 (Remarks to the Author):

Overall, this manuscript describes the role of changes in N and P supply on the timely and significant increase in development of Sargassum macroalgal blooms from more limited populations in the Sargasso Sea and Gulf of Mexico during the mid-late 1900s to the recent expansion across the S. Atlantic Ocean from West Africa to Brazil and northward into the Caribbean waters. This massive expansion in algal biomass has recently been tracked through satellite imagery, and is continuing to grow in size. To feed these algal blooms, there is no doubt that multiple sources and inputs of

nutrients must be available along their track to sustain such high biomass.

This study takes baseline measurements of nutrient content of the algae from the 1980s and compares that to recent samples taken from the region, taking into account changes in the overall nutrient content of N and P and the stoichiometry over time. The methodologies used are simple techniques that have been shown to be great indicators of N and P limitation and of changes in eutrophication status of coastal areas. However, despite their simplicity, when used over time and across large regions, they can accurately show large scale patterns in nutrient limitation and identify locations of high nutrient inputs that may need stricter management for mitigation to be successful. Overall I find this paper timely quite important to better understand what is driving the largest algal bloom worldwide.

I find the introduction to be nicely written for a broader audience with interesting historical references, bringing together some of important current understandings of the recent increase in Sargassum expansion across the Atlantic, Caribbean, and Sargasso Sea. Only a few minor edits:

Line 54 can you add a short definition to explain what you mean by "windrow" here for the broader reader

Line 55 should read "1980s have led"

Line 61 I think you mean to refer to Ryther's paradox cited from reference 1. I would suggest to add the citation here again.

Line 71 remove "the" before 2018.

Line 81 should read "and a number of suitable"

Lines 99-102- I find this objective a bit misleading. Although you have samples from 1980s and 2010 which span 40 years, you are not actually looking over those four decades but rather a snap shot of the past in comparison to today. That would be one critique of the study that there is no continuous or decadal sampling that shows the development overtime. It is highly likely that a stronger quantitative temporal pattern would be more dynamic. I don't know whether this would be possible to change if no other samples exist in between these time points. Otherwise, I would suggest to specify more clearly that you are looking at 40 years in the past and present but not during the past four decades.

Results:

Here I would start first with the overview of the nutrient data bringing up lines 117-132. This is the more important message you are trying to make. I think the similarities between species (Lines 106-115) is less significant to the overall story and should be only briefly mentioned in the main text and rather included in the Supplemental data. Also, here you first mention that Looe Key is a long-term data set. If that is so and more data exist over time, then you need to specify that above in your objective that you provide a site-specific example to look at the trend over time.

In line 117, you also mention the change in tissue over broad areas GOM, NA, and Caribbean Sea, however Fig. 2 seems to be only for NA dataset. Does that include the three sub-regions? Please specific.

Figure 3 shows 1980s to 2010 in Looe Key reef. If you have a long-term dataset here with multiple time points, I would suggest that you use it better. As is, this figure is repetitive to Fig. 2 and does not show anything different. However, you may be able to better use this example to show the trend over time in N inputs and P limitation and possibly a rate of this increase. This would really strengthen the dataset. Otherwise I would remove the figure and stick with the broader patterns NA region. The seasonal differences are slightly different between the NA region and Looe Key reef, but this is rather

a side story rather than the take home message you want to push through.

Figure 4. Here I would suggest using a map to show these values for each site (with size of circle representing the mean values). Although most sites are from 2010, you could add the 1980 data as well where you have it. This would allow for a larger spatial analysis to see where the hotspots are located (i.e. coastal areas with high riverine inputs).

How did you determine % changes over time in areas where no 1980s data existed? Make sure that is clear as well (not just in the results, but perhaps a short statement in the results to indicate that it is compared to the average 1980s value, etc.

Discussion:

Line 238-253, It is mentioned that sargassum growth in the Sargasso Sea used primarily recycled nutrients from organic matter and ammonium-rich excretions, whereas more recently in the coastal areas NO_x is the main source of N. Are there consequences of this shift in source of N as not all algae take up and assimilate the different N sources the same way. It would be nice to discuss uptake rates among these sources if these data are available for these species.

Lines 255-269-If lower salinity waters are driving high biomass production, is there any evidence that this species has a high tolerance for changes in salinity? Has it been found also growing in coastal bays or even estuarine systems or only in the open ocean? If they are able to cope with lower salinities, this could have further coastal implications. Please discuss.

In your discussion you point out the number of large sources of nutrients, ie Amazon, Mississippi, dust, upwelling, as the most likely sources of nutrients driving the blooms. I would like to know whether in the Caribbean where there are large biomasses washing ashore in which there are also touristic areas, coral reefs, and often blooms remain for extended periods, whether local non-point sources could also be playing a role to sustain blooms longer, or whether these sources are rather insignificant in the larger picture. I think this would be an interesting aspect to study further as although it may not be the likely nutrient source driving the blooms, it may provide a better understanding of mitigation measures that could reduce the local impacts.

NCOMMS-20-42996

Nutrient content and stoichiometry of pelagic *Sargassum* reflects increasing nitrogen availability in the Atlantic Basin

B. E. Lapointe, R. A. Brewton, L. W. Herren, M. Wang, C. Hu, D. J. McGillicuddy, Jr., S. Lindell, F. J. Hernandez, P. L. Morton

REVIEWER COMMENTS

Reviewer #1 (Remarks to the Author):

The authors have conducted an interesting study determining the effects of inorganic nitrogen availability on *Sargassum* sp. blooms in waters of the North Atlantic Ocean. They have found that the occurrence of these blooms are strictly associated with changes in inorganic nitrogen and phosphorus inputs from the last decades (terrestrial runoff and river discharges). The manuscript is very well written and easy to follow, but there a few points that I would like to pointed out.

Inorganic nitrogen and phosphorus play important roles in macroalgae physiology and ecology, limiting their growth. The current study have shown a similar pattern in the N:P ratio from river discharges and the N:P ratio in algal tissue, showing that this ratio have significantly increased in the past decades likely due to eutrophication. *Sargassum* sp. usually grows in oligotrophic waters, and hence it is particularly interesting evaluate the impact of eutrophication on their abundances and distribution patterns as their blooms have caused great ecological and social impacts in the North Atlantic ocean. Overall, I think this is a very nice study, and it will be of great importance for decision-making authorities.

Reply: Thank you for the positive comments!

One of my suggestions would be to include more details about the collection sites in the introduction and/or map (Fig.1) because the method section is at the end of the ms, and sometimes it is difficult to follow the result section. In addition, I saw in the method section that the authors have also measured $\delta^{13}\text{C}$ in the samples, why these results were not include or discussed through the ms? Are they relevant for the study or not?

Reply: Thanks for pointing the need to describe collection sites earlier in the manuscript. We have moved these details from the Methods up to the beginning of the Results and agree they are a better fit here. Secondly, as this study focuses on the role of nitrogen, the carbon isotopes were not an important or relevant discussion point. Therefore, we have removed the mention of $\delta^{13}\text{C}$ from the Methods and the data Supplemental Table 3 to clarify this.

On the discussion line 234 the authors mentioned that previous studies have measured N and P content in the same species. How these values are compare to those obtained in the current study? I could see the raw data in the Supplementary table S4, and they are quite low, except by the summer season where they are influenced by the river discharges. So, It would be nice to see a short paragraph describing or comparing these results with previous studies. This species was usually exposed to oligotrophic waters, and now it exposed to very euthropied waters, so it will be nice to compare the N and P content to see if they are capable to store it. Do you think they have change their growth strategy? Why limiting phosphorus does not limit their growth?

Reply: The current paper specifically compared the *Sargassum* tissue N and P data from our previous studies in the 1980s (Lapointe 1995; Lapointe et al. 2014) to more recent measurements since 2010. These are the 1980s data shown in Fig. 2, Fig. 3 (previously Fig. 4), Supplemental Fig. 1, Supplemental Fig. 2, and Supplemental Table 3 We have clarified this comparison at the end of the Introduction and in the Discussion by citing the previous papers, as well as in the Fig. 2 legend.

The lead author conducted experimental growth assays with *Sargassum* in the 1980s and demonstrated enhanced growth with enrichment of either N or P (Lapointe 1987, 1995). Potential changes in growth strategy are very interesting, but beyond the scope of this study.

Line 288: delete soluble reactive phosphorus (described above).

Reply: Thanks for the careful reading! It is now deleted as suggested.

Line 322: how could you describe the growth pattern in this species, anticipator or responder?

Can its growth strategies change across their distribution range?

Reply: Very interesting question. Pelagic *Sargassum* appears to be a responder rather than an anticipator because its growth rate and the productivity responds positively to increased nitrogen availability. Accordingly, it is similar to temperate kelps such as *Laminaria longicuris* and *Macrocystis pyrifera* that also respond to increased nitrogen with increased growth (Kain 1989). Our previous research (Lapointe et al. 2014) showed that productivity and growth both increased in *Sargassum* with increasing nutrients over broad areas of its distribution (Caribbean, Gulf Stream, Sargasso Sea). We have added to this paragraph in the text and included the citation to Kain (1989) for the concept of anticipator vs. responder.

Lines 321-325: how these attributes contribute to their nitrogen physiology (or nitrogen uptake strategies)? It is not clear how its air bladders or reproduction patterns can favour their distribution in nutrient-enriched waters.

Reply: The air bladders allow *Sargassum* to float and access buoyant river plumes at the ocean surface. We have clarified this concept and removed the mention of reproductive patterns for brevity.

Reviewer #2 (Remarks to the Author):

Nature Communications MS NCOMMS--20--42996

Nutrient content and stoichiometry of pelagic *Sargassum* reflects increasing nitrogen availability in the Atlantic Basin

By B. E. Lapointe, R. A. Brewton, L. W. Herren, M. Wang, C. Hu, D. J. McGillicuddy, Jr., S. Lindell, F. J. Hernandez, P. L. Morton

General

Lapointe et al. have a priceless data set from 40 years ago to compare to a current nutrient content, nutrient type and isotope signature data set on pelagic *Sargassum* in the Gulf of Mexico, Florida Keys and open Atlantic waters. They make a convincing argument that N input through Mississippi River discharge and land sources to the GOM has significantly increased with global growth of artificial fertilizers and contributed to increased N content of *Sargassum* there. They do not show a quantitative increase in the biomass of *Sargassum* in the GOM and that would a useful addition

to the argument in this paper. Their speculation that this increase in fertilization is also happening in the GASB is in line with the speculation of many other authors. However, the data base provided for this speculation is much smaller, the values much lower and the story weaker than the one for the GOM. The authors are correct to indicate more research is needed. This paper with improvements is a useful contribution to the ongoing puzzle of why *Sargassum* has increased so much in the GASB but does not yet solve the puzzle.

Reply: We appreciate the “priceless” comment! We also agree that the linkage between Mississippi River discharges and N input to the GOM and *Sargassum* is convincing. Measuring a quantitative increase in *Sargassum* from winter to summer when the river discharges increase is beyond the scope of this manuscript; however, increased *Sargassum* biomass coinciding with increasing Mississippi River discharges has already been established by Gower and King (2008) who used MERIS satellite imagery and concluded “our observations show a large increase in *Sargassum* in the northwest GOM between March and June of each year.” We have added this to the text and clarified the linkage. We agree that such a connection to the Amazon, Congo, and Orinoco rivers is weaker, hence our conclusion that more research is needed. An additional figure of remotely sensed *Sargassum* biomass data showing this increase in the GOM has been added to the Supplement (Supplemental Fig. 3) and mentioned in the text.

Specific

P1 line 40 might be first identified at “Ryther’s” paradox.

Reply: Very nice suggestion, this change has been made.

P6 Results No actual results on a quantitative increase in *Sargassum* in the GOM and export to the NA is presented. It would be a useful contribution to include such data analogous to Figure 6d.

Reply: Thank you for the suggestion. We have cited Gower and King (2008) who used MERIS to show the seasonal export from GOM to NA and have also added a figure to the Supplement showing this pattern (Supplemental Fig. 3) above.

Figure 4abc Useful to indicate statistical significance of differences along the gradient.

Reply: Although there are significant statistical differences for %N and N:P if you compare the 1980s average to each bar on the plots, we felt that as these same data were already analyzed in the overall decadal statistics (Fig. 2 and Supplemental Tables 1 and 2) it was more confusing and complicated to add on additional statistics here. Further, some of these points represent very small sample sizes and the distribution is unequal across the locations, thus unfortunately, while significance was observed that supported our observations, we felt that this statistical analysis was not very sound, even using a non-parametric test. However, we have added dotted lines indicating critical values on these barplots (See Fig. 3 legend) that will help the reader understand the values. The primary purpose of this figure was to highlight the geographical distribution of the data, so we have also included a map version of these same data in the Supplement (Supplemental Fig. 2).

Figure 4 b Useful to add line of global N:P ratio of 28 onto graph?

Reply: Thank you for the suggestion. This change has been made on what is now Fig. 3.

Figure 5a What is SMOS? What is the source of 2018 information on the distribution of *Sargassum*?

Reply: Thanks for catching this, we have now added the full name of SMOS (Soil Moisture and Ocean Salinity (SMOS) mission) and the 2018 *Sargassum* distribution data source in the figure caption (Wang et al., 2019).

Figure 5b Salinity normalized to what? What is the raw data abundance and distribution for what year(s)?

Reply: In the caption for Fig. 4b (previously Fig. 5b), we have specified that data analyzed were from 2011 to 2019 and included more detailed descriptions on frequency distribution plots as follows: “Here, the raw data (purple) shows *Sargassum* areal coverage in each salinity increment, relative to the total coverage (i.e., all purple bars sum to 1.0). The *Sargassum* coverage in each salinity increment is divided by the water area in the same salinity increment, resulting in the “Salinity normalized” distribution (blue). **The *Sargassum* distribution data were obtained on National Centers for Environmental Information (NCEI) Accession 0190272¹⁴.**”

P15 line 257-259 explain calculation and abundance of *Sargassum* for what year(s).

Reply: The explanation of the statistics is revised as: “The statistical analysis of the *Sargassum* cover in different salinity ranges during the time period of 2011 - 2019 shows that ...”

Figure 5c What do the colors over land indicate -average area burned. Perhaps we need a key?

Reply: Thank you for the suggestion. This change has been made on Fig. 4c (previously Fig. 5c).

Figure 6abc Need reference for CAMREX cruises. It would be nice to show regressions for post 2010 data on nutrient discharge?

Reply: We have now added the reference in the figure caption (now Fig. 5). All the references for these data sources are also summarized in the Method under: “Satellite-measured distributions of *Sargassum* and salinity and field-measured river nutrient concentrations”.

For the post 2010 data on nutrient discharge, no data were collected for the CAMREX cruises so here we analyzed the HYBAM dataset for the post 2010 data (see Fig. R1 in the right panel). The patterns are similar to the results derived for the HYBAM dataset from 2003 to 2018 (see Fig. R1 in the left panel): both showing significant positive correlations between nutrient flux and river discharge.

Figure R1. Nitrate flux at Obidos from the HYBAM database (Cochonneau et al., 2006) is correlated with discharge, and the correlation is higher if data are binned to different discharge groups. The plot on the right was generated using the HYBAM data from 2003 to 2018, while the plot on the left only includes the HYBAM data from 2011 to 2018, both showing similar significant correlation between nutrient flux and river discharge.

Figure 6d Need reference for HYBAM data set. Looks like max value for nitrate Y-axis could be 2000 kg/s. Why is there so little data for most years compared to 2003-4?

Reply: Cochonneau et al. (2006), which described the sample analyzing procedures for the HYBAM dataset has been added as a reference here. Recently **HYBAM dataset has just been updated** (applied some quality controls and additional processing and included more recent data). Therefore, we have updated Fig. 5c (previously Fig. 6d) with the new HYBAM dataset as shown below.

Figure R2. Nitrate flux at Obidos from the same HYBAM dataset shows apparent increases in recent years. The recent Sargassum areal coverages obtained from Wang et al., 2019 in the Central West Atlantic (CWA, 0oN – 22oN, 63oW – 38oW) are also shown as reference. **The shaded gray**

bars indicate when quality-controlled with HYBAM nutrient flux measurements are available.

For **the high nitrate flux** (~1500 kg/s, such as in 2014 and 2016), those cases are typically associated with high nitrate concentration of ~6 mg/L and discharge of ~220000 m³/s.

For **the data gaps of HYBAM data**, Fig. R2 above shows that in most years (except for 2008-2009 and 2014-2015), measurements are available (see the shaded gray bars) in nearly all months, but data in some months are simply **too low (usually less than 0.1 mg/L)** to show up.

Provide reference for the source of the *Sargassum* data for the CWA. Data for 2018 shown for Figure 5 not included here?

Reply: We have now revised to include the *Sargassum* data source as follows: “The recent *Sargassum* areal coverages obtained from Wang et al., 2019 in the Central West Atlantic ...”.

In the previous version, the 2018 data was not shown because there was no 2018 nutrient data available on HYBAM at that time. We have now updated Fig. 5d (previously Fig. 6d) and included the 2018 data for both nitrate flux from HYBAM and *Sargassum* coverage from satellite observations. Note that **the *Sargassum* time series in Fig. 5c (previously Fig. 6d) is also revised** (in the previous plot the area coverage was not estimated for the exact CWA geographical range of 0°N – 22°N, 63°W – 38°W).

P17 line 299 Location of 38°W should be noted on Figure 5a. Equatorial counter current in early summer may carry the Amazon River plume further offshore and to Africa? See distribution of lower salinities on Figure 5a.

Reply: We have added a blue line on Fig. 4a (previously Fig 5a) to indicate the location of 38°W. Although the equatorial counter currents can reach waters east of 38°W, Amazon River plume is mostly restricted to the west of 38°W as shown in the salinity maps.

P20 Sources and signatures of N-15 should usefully include Sargasso Sea, sewage, CWA, Congo and Amazon River discharge data.

Reply: Unfortunately, we do not have $\delta^{15}\text{N}$ data for *Sargassum* in the Sargasso Sea. However, we have included in the Discussion $\delta^{15}\text{N}$ source data for sewage, synthetic fertilizers, atmospheric deposition, upwelled nitrate, Mississippi River discharges, and published values for macrophytes and phytoplankton in the Amazon basin published by Zaia Alves et al. (2017), which was added to the references.

P20 line 354-364 Do you suggest that N₂ fixation has increased in GASB in correlation with the bloom increases of *Sargassum*? If so it would be useful to provide further evidence of the reason for such an increase in N₂ fixation such as greater Fe input to the area.

Reply: Thank you for the suggestion. The relationship between increased N₂ fixation and Fe deposition from Africa was added to the paragraph below line 354-364 on atmospheric nutrient inputs.

P21 line 368-382 If there is any evidence that this aerosol source of nutrients has increased in correlation with the bloom increases of *Sargassum* in the GASB it would be useful to include it here.

Reply: We have revised this paragraph to include information on increased aerosols with additional references.

P24 line 450 define NCEI.

Reply: We have added the full name (National Centers for Environmental Information) and defined “NCEI” in the figure caption of Fig. 4b (previously Fig. 5b).

Candace Oviatt

Reviewer #3 (Remarks to the Author):

Overall, this manuscript describes the role of changes in N and P supply on the timely and significant increase in development of *Sargassum* macroalgal blooms from more limited populations in the Sargasso Sea and Gulf of Mexico during the mid-late 1900s to the recent expansion across the S. Atlantic Ocean from West Africa to Brazil and northward into the Caribbean waters. This massive expansion in algal biomass has recently been tracked through satellite imagery, and is continuing to grow in size. To feed these algal blooms, there is no doubt that multiple sources and inputs of nutrients must be available along their track to sustain such high biomass.

This study takes baseline measurements of nutrient content of the algae from the 1980s and compares that to recent samples taken from the region, taking into account changes in the overall nutrient content of N and P and the stoichiometry over time. The methodologies used are simple techniques that have been shown to be great indicators of N and P limitation and of changes in eutrophication status of coastal areas. However, despite their simplicity, when used over time and across large regions, they can accurately show large scale patterns in nutrient limitation and identify locations of high nutrient inputs that may need stricter management for mitigation to be successful. Overall I find this paper timely quite important to better understand what is driving the largest algal bloom worldwide.

Reply: Thank you for the nice summary and positive comments!

I find the introduction to be nicely written for a broader audience with interesting historical references, bringing together some of important current understandings of the recent increase in *Sargassum* expansion across the Atlantic, Caribbean, and Sargasso Sea.

Reply: Thank you again.

Only a few **minor edits**:

Line 54 can you add a short definition to explain what you mean by “windrow” here for the broader reader

Reply: Revised as suggested.

Line 55 should read “1980s have led”

Reply: Thanks for the careful reading! It is now revised as suggested.

Line 61 I think you mean to refer to Ryther’s paradox cited from reference 1. I would suggest to add the citation here again.

Reply: Revised as suggested.

Line 71 remove “the” before 2018.

Reply: Revised as suggested.

Line 81 should read “and a number of suitable”

Reply: Revised as suggested.

Lines 99-102- I find this objective a bit misleading. Although you have samples from 1980s and 2010 which span 40 years, you are not actually looking over those four decades but rather a snapshot of the past in comparison to today. That would be one critique of the study that there is no continuous or decadal sampling that shows the development overtime. It is highly likely that a stronger quantitative temporal pattern would be more dynamic. I don't know whether this would be possible to change if no other samples exist in between these time points. Otherwise, I would suggest to specify more clearly that you are looking at 40 years in the past and present but not during the past four decades.

Reply: Thank you for the input. We agree that as it was written this did not represent the study well. Unfortunately, there are no data points during the “interim.” So, we have removed the phrase “during the past four decades” to clarify the objective of the study. The final sentence reads as follows, “Here, the objective is to better understand the effects of N and P supply on *Sargassum* ~~during the past four decades~~, where a unique baseline *Sargassum* tissue C:N:P data set from the 1980s are compared with more recent samples collected since 2010 (Fig. 1).”

Results:

Here I would start first with the overview of the nutrient data bringing up lines 117-132. This is the more important message you are trying to make. I think the similarities between species (Lines 106-115) is less significant to the overall story and should be only briefly mentioned in the main text and rather included in the Supplemental data. Also, here you first mention that Looe Key is a long-term data set. If that is so and more data exist over time, then you need to specify that above in your objective that you provide a site-specific example to look at the trend over time.

Reply: Thanks for the suggestion. We have moved the discussion by species to the Supplement. The additional Looe Key objective has also been removed completely for brevity as per the later suggestion.

In line 117, you also mention the change in tissue over broad areas GOM, NA, and Caribbean Sea, however Fig. 2 seems to be only for NA dataset. Does that include the three sub-regions? Please specific.

Reply: Thank you for pointing out the need for this clarification. Our use of the term NA includes the GOM and Caribbean Sea and this has been clarified globally throughout the text, particularly on lines 117-119.

Figure 3 shows 1980s to 2010 in Looe Key reef. If you have a long-term dataset here with multiple time points, I would suggest that you use it better. As is, this figure is repetitive to Fig. 2 and does not show anything different. However, you may be able to better use this example to show the trend over time in N inputs and P limitation and possibly a rate of this increase. This would really strengthen the dataset. Otherwise I would remove the figure and stick with the broader patterns

NA region. The seasonal differences are slightly different between the NA region and Looe Key reef, but this is rather a side story rather than the take home message you want to push through.

Reply: This is a good suggestion. Because the Looe Key data are included in the overall NA dataset shown in Fig. 2, we agree it is best to remove the separate Looe Key figure and analyses to simplify and shorten the manuscript.

Figure 4. Here I would suggest using a map to show these values for each site (with size of circle representing the mean values). Although most sites are from 2010, you could add the 1980 data as well where you have it. This would allow for a larger spatial analysis to see where the hotspots are located (i.e. coastal areas with high riverine inputs).

Reply: Thank you. This is another good suggestion and we have included a map figure of these data in the supplement (Supplemental Fig. 2). We are happy to include whichever the editor prefers in the manuscript.

How did you determine % changes over time in areas where no 1980s data existed? Make sure that is clear as well (not just in the results, but perhaps a short statement in the results to indicate that it is compared to the average 1980s value, etc).

Reply: Sorry for the confusion. Fig. 3a and b (previously Fig. 4a and b) show the %N and N:P from 2010 samples, not percent changes since the 1980s. The coloring represented the locations where the 2010s were greater than the 1980s overall averages. We have revised and clarified this in the caption by rewording. The caption now reads as follows, “Fig. 3: Post-2010 *Sargassum* tissue properties by location, as well as Northern Hemisphere meteorological season for Gulf of Mexico (GOM) samples, indicating where %N and N:P ratios were greater than the 1980s baseline average for the entire dataset (black dotted lines). a) For %N, values have significantly increased from the 1980s (decadal average = 0.89%) to post-2010 (decadal average = 1.21%); %N values > 1.5 (red dashed line) are considered non-limiting to growth³⁷. b) N:P ratios have significantly (111%; ANOVA, $F = 193.4$, $P < 0.001$) increased from the 1980s (decadal average = 13.2) to post-2010 (decadal average = 27.8, blue dashed line). c) Enriched $\delta^{15}\text{N}$ values (>3‰, orange dotted line) are indicative of urbanized wastewater discharges; while more depleted values are indicative of N_2 fixation, atmospheric deposition, and upwelling.”

Discussion:

Line 238-253, It is mentioned that *sargassum* growth in the Sargasso Sea used primarily recycled nutrients from organic matter and ammonium-rich excretions, whereas more recently in the coastal areas NO_x is the main source of N. Are there consequences of this shift in source of N as not all algae take up and assimilate the different N sources the same way. It would be nice to discuss uptake rates among these sources if these data are available for these species.

Reply: Han et al. (2018) measured uptake rates of ammonium, nitrate, and urea for the benthic species *Sargassum hemiphyllum*. The data show that ammonium uptake capacity by *Sargassum* is greater than nitrate and urea (like many macroalgae), but *Sargassum* can assimilate all three forms. We have added this detail to the text and cited Han et al. (2018).

Lines 255-269-If lower salinity waters are driving high biomass production, is there any evidence that this species has a high tolerance for changes in salinity? Has it been found also growing in coastal bays or even estuarine systems or only in the open ocean? If they are able to cope with lower salinities, this could have further coastal implications. Please discuss.

Reply: This is a valid point. Currently, studies on the impact of salinity on *Sargassum* growth are very scarce. Below is a figure from Hanisak and Samuel (1987) showing the growth rate of *S. natans* at different salinity levels.

The figure suggests that *Sargassum* growth rate is actually lower at low salinity, and growth rate decreases to 0 for salinity ≤ 12 . Therefore, although river discharge provides nutrients to stimulate *Sargassum* growth, low salinity (≤ 30) may be a limiting factor.

In your discussion you point out the number of large sources of nutrients, ie Amazon, Mississippi, dust, upwelling, as the most likely sources of nutrients driving the blooms. I would like to know whether in the Caribbean where there are large biomasses washing ashore in which there are also touristic areas, coral reefs, and often blooms remain for extended periods, whether local non-point sources could also be playing a role to sustain blooms longer, or whether these sources are rather insignificant in the larger picture. I think this would be an interesting aspect to study further as although it may not be the likely nutrient source driving the blooms, it may provide a better understanding of mitigation measures that could reduce the local impacts.

Reply: Very good point. We have added the relationship between human activities in Florida and Texas with elevated $\delta^{15}\text{N}$ and N:P ratios in *Sargassum* to the Discussion; indeed, this phenomenon would also be occurring on smaller scales throughout tourist areas of the Caribbean, such as Yucatan, Mexico.

References

Cochonneau, G. et al. The Environmental Observation and Research project, ORE HYBAM, and the rivers of the Amazon basin. The Fifth FRIEND World Conference held Climate: Variability and Change— Hydrological Impacts **308**, (2006).

Han, T., Qi, Z., Huang, H., Liao, X., & Zhang, W. Nitrogen uptake and growth responses of seedlings of the brown seaweed *Sargassum hemiphyllum* under controlled culture conditions. *J. Appl. Phycol.* **30**, 507-515 (2018).

Hanisak, M. D., & Samuel, M. A. Growth rates in culture of several species of *Sargassum* from Florida, USA. *In Twelfth International Seaweed Symposium* (399-404). Springer, Dordrecht. (1987).

Kain, J. M. The seasons in the subtidal. *Br. Phycol. J.* **24**, 203-215 (1989).

Lapointe, B. E., Littler, M. M., & Littler, D. S. A comparison of nutrient-limited productivity in macroalgae from a Caribbean barrier reef and from a mangrove ecosystem. *Aq. Bot.* **28**(3-4), 243-255 (1987).

Lapointe, B. E. A comparison of nutrient-limited productivity in *Sargassum natans* from neritic vs. oceanic waters of the western North Atlantic Ocean. *Limnol. Ocean.* **40**(3), 625-633(1995).

Lapointe, B. E., West, L. E., Sutton, T. T., & Hu, C. Ryther revisited: nutrient excretions by fishes enhance productivity of pelagic *Sargassum* in the western North Atlantic Ocean. *J. Exp. Mar. Biol. Ecol.* **458**, 46-56 (2014).

Wang, M. et al. The great Atlantic *Sargassum* belt. *Science.* **364**, 83–87 (2019).

REVIEWERS' COMMENTS

Reviewer #1 (Remarks to the Author):

The authors have addressed most (or all) of the questions and suggestions made by the reviewers, improving the manuscript fluency and content. I have only minor changes before to go for acceptance.
On line 141: Is it ok to mention "by decade" when you have the years 1980 and 2010 in your graph?
On line 195: change *Sargassum* to italic.
On lines 334-351: You mentioned most of the N15 signatures for different conditions (e.g., urban wastewater, synthetic fertilizer N, etc.), but what about the N15 signature for natural oceanic ammonium and nitrate sources?

Pamela A. Fernández

Reviewer #2 (Remarks to the Author):

Dear Editors,
I am satisfied with the changes the authors have made to the manuscript according to my earlier review comments. I find the manuscript acceptable and an useful addition to the literature on sargassum blooms.

Reviewer #3 (Remarks to the Author):

I appreciate the consideration of my previous comments by the authors in this revised version of the manuscript. As suggested by the authors, I also would accept the decision of the editors as to whether to use the Supplementary Figure of the Map in the main text or leave it as is in the Supplement. I do think it helps to visualize the data more clearly in relation to the river plumes, but it may be repetitive to have two maps in the main text along with the site locations.

Overall, I believe that with the additional revisions that I and the other reviewers suggested, that the paper has improved in its communication of the data, is clearer and more concise. I have no further changes to the current version. I believe it will make a great contribution to the field, is timely, and will drive further research on this topic to better understand the role nutrients play in driving these algal blooms.

Mirta Teichberg

NCOMMS-20-42996

Nutrient content and stoichiometry of pelagic *Sargassum* reflects increasing nitrogen availability in the Atlantic Basin

B. E. Lapointe, R. A. Brewton, L. W. Herren, M. Wang, C. Hu, D. J. McGillicuddy, Jr., S. Lindell, F. J. Hernandez, P. L. Morton

Response to Reviewers 2

REVIEWERS' COMMENTS

Reviewer #1 (Remarks to the Author):

The authors have addressed most (or all) of the questions and suggestions made by the reviewers, improving the manuscript fluency and content. I have only minor changes before to go for acceptance.

We thank the reviewer for their careful reading and thoughtful comments.

On line 141: Is it ok to mention “by decade” when you have the years 1980 and 2010 in your graph?

The “by decade” refers to the 1980s vs 2010s shown in the barplot of panel (a). The second part of the figure panel (b) shows the data by meteorological season. We feel that removing this phase might be confusing to the reader. If the editor feels the phrase needs to be removed, we are open to that suggestion. This was also clarified on line 126 in the Results. “Elemental composition varied significantly ~~by decade~~ **between these two decades.**”

On line 195: change *Sargassum* to italic.

Correction made.

On lines 334-351: You mentioned most of the N15 signatures for different conditions (e.g., urban wastewater, synthetic fertilizer N, etc.), but what about the N15 signature for natural oceanic ammonium and nitrate sources?

Additional references and discussion relating to natural oceanic $\delta^{15}\text{N}$ values has been included and this section now reads as follows, “**Similarly, in Bermuda, rainwater NH_4^+ $\delta^{15}\text{N}$ values ranged from -12.5 to +0.7 ‰¹** Synthetic fertilizer N, which has increased nine-fold since the

1980s, has $\delta^{15}\text{N}$ values ranging from -2 to +2 ‰ and is the mid-range for most *Sargassum* values in this study. More enriched values of +2.5 to +4.8 ‰ are indicative of upwelled NO_3^- in the upper 200 m of the NA^{2,3}.”

Pamela A. Fernández

Reviewer #2 (Remarks to the Author):

Dear Editors,

I am satisfied with the changes the authors have made to the manuscript according to my earlier review comments. I find the manuscript acceptable and an useful addition to the literature on sargassum blooms.

Thank you to Reviewer #2 for their positive feedback.

Reviewer #3 (Remarks to the Author):

I appreciate the consideration of my previous comments by the authors in this revised version of the manuscript. As suggested by the authors, I also would accept the decision of the editors as to whether to use the Supplementary Figure of the Map in the main text or leave it as is in the Supplement. I do think it helps to visualize the data more clearly in relation to the river plumes, but it may be repetitive to have two maps in the main text along with the site locations.

We agree that this decision can be left to the editor.

Overall, I believe that with the additional revisions that I and the other reviewers suggested, that the paper has improved in its communication of the data, is clearer and more concise. I have no further changes to the current version. I believe it will make a great contribution to the field, is timely, and will drive further research on this topic to better understand the role nutrients play in driving these algal blooms.

Thank you to Reviewer #3 for the positive feedback.

Mirta Teichberg

New References:

1. Altieri, K. E., Hastings, M. G., Peters, A. J., Oleynik, S. & Sigman, D. M. Isotopic evidence for a marine ammonium source in rainwater at Bermuda. *Global Biogeochem. Cycles* **28**, 1066-1080 (2014).
2. Knapp, A. N., DiFiore, P. J., Deutsch, C., Sigman, D. M. & Lipschultz, F. Nitrate isotopic composition between Bermuda and Puerto Rico: Implications for N₂ fixation in the Atlantic Ocean. *Global Biogeochem. Cycles* **22** (2008).
3. Knapp, A. N., Sigman, D. M., Lipschultz, F., Kustka, A. B. & Capone, D. G. Interbasin isotopic correspondence between upper-ocean bulk DON and subsurface nitrate and its implications for marine nitrogen cycling. *Global Biogeochem. Cycles* **25** (2011).